# Community-Driven Insights into Fish Assemblage, Microhabitats, and Management Strategies in the Meghna River Basin of Bangladesh

**Mst. Armina Sultana** [1,*] **, Md. Ashraf Hussain** [2] **, Petra Schneider** [3,*] **, Md. Nahiduzzaman** [4] **, Benoy Kumar Barman** [4] **, Md. Abdul Wahab** [4,5] **, Mohammad Mojibul Hoque Mozumder** [6] **and Mrityunjoy Kunda** [1]

1   Department of Aquatic Resource Management, Sylhet Agricultural University, Sylhet 3100, Bangladesh; kunda.arm@sau.ac.bd
2   Department of Fisheries Technology and Quality Control, Sylhet Agricultural University, Sylhet 3100, Bangladesh; ashraf.ftqc@sau.ac.bd
3   Department for Water, Environment, Civil Engineering and Safety, University of Applied Sciences Magdeburg-Stendal, Breitscheidstraße 2, D-39114 Magdeburg, Germany
4   WorldFish Bangladesh, South Asia Office, Dhaka 1213, Bangladesh; m.nahiduzzaman@cgiar.org (M.N.); b.barman@cgiar.org (B.K.B.); wahabma_bau2@yahoo.com (M.A.W.)
5   Center for Blue Resources Development—CBRD, Dhaka 1229, Bangladesh
6   Fisheries and Environmental Management Group, Helsinki Institute of Sustainability Science (HELSUS), Faculty of Biological and Environmental Sciences, University of Helsinki, 0014 Helsinki, Finland; mohammad.mozumder@helsinki.fi
*   Correspondence: armina.arm@sau.ac.bd (M.A.S.); petra.schneider@h2.de (P.S.)

**Abstract:** The present study aimed to delve into the local ecological knowledge of fisheries in the Meghna River Basin (MRB) of Bangladesh by exploring the insights and perspectives of local communities. A survey was administered among six fishing communities from five districts along the MRB between August 2015 and January 2016 to accumulate data for this study. The study sites were selected meticulously based on three crucial criteria: upstream river, coastal area, and fish sanctuaries, which covered three major rivers, namely the Meghna, Andharmanik, and Payra. The study employed participatory rural appraisal (PRA) tools, including 120 individual interviews using purposive sampling, 25 focus group discussions, and 36 key informant interviews. The study identified the ten most frequently caught fish species along with their temporal and spatial variation as reported by the respondents. Nine of these species fell into the least concern category, which indicate their stable population status. Meanwhile, six out of ten species cited as highly caught in the previous one to two decades belong to the threatened or near-threatened category. Findings also reveal that fishers are able to recognize important microhabitats of the study area and their significance for fish species. In addition, fishers identified the negative drivers of ecosystem degradation as well as suggested several management measures to address these challenges. The results of this study underscore the critical role of engaging with local communities and integrating their ecological knowledge into initiatives for the sustainable exploitation and conservation of aquatic resources in the MRB.

**Keywords:** local ecological knowledge; fisheries; fishing practices; ecological impacts; sustainable exploitation

## 1. Introduction

Bangladesh, situated within the Ganges, Brahmaputra, and Meghna (GMB) river systems, benefits from its deltaic location with abundant aquatic biodiversity and significant fisheries potential [1]. Fish is one of the most important sources of animal protein, with the average person in Bangladesh consuming more than 22 kg of fish annually [2]. Inland

capture fisheries are a significant source of food and livelihood for the people of Bangladesh, with an estimated annual production of over 1.25 million metric tons of fish, making it the world's third-largest producer [3]. Additionally, fishing and related activities are a crucial part of the livelihoods of many coastal and riverine communities in Bangladesh, providing employment and income for millions of people.

The Meghna River Basin (MRB) is one of the largest inland depressions and the richest wetland [4]. A large number of catadromous, anadromous, and diadromous fishes use the basin for breeding, feeding, and migratory purposes [5]. Many commercially important fish species, especially *Tenualosa ilisha*, depend on coastal rivers for spawning and nursery grounds [6]. The aquatic ecosystem is of immense ecological, social, and economic importance to the country, providing vital ecosystem services, such as water supply, nutrient cycling, and habitat provision, as well as supporting numerous livelihoods, particularly those dependent on fisheries [7].

Since rivers are a dynamic system due to their hydrology and morphology, some changes occur in slopes, dikes, and riverine islands depending on their energy [8]. However, overfishing, anthropogenic pressure, unplanned flood control, and deficiencies in irrigation infrastructures also lead to the destruction of river ecosystems [9]. Especially microhabitats, where fish spend all or part of their time avoiding predatory species or reducing interaction with competitors in competition for food, are affected by these changes [10]. Convenient fish habitat is a crucial factor for the robustness and sustainability of fish populations [11]. Nevertheless, destroying the habitats over time has resulted in habitat declination for many fish species by leaving their natural habitats and even the disappearance of breeding, feeding, and nursery areas [12]. In addition, IUCN Bangladesh categorized a total of 64 freshwater fish species as threatened, with 9 critically endangered, 30 endangered, and 25 vulnerable among them [13]. Therefore, it is of great importance to conduct a scientific evaluation of the effects of natural and human activities on the fish biodiversity of the MRB. Such an assessment would serve as a guide for the development of effective management strategies.

Indigenous knowledge is a type of community-driven knowledge that is passed down through generations and is specific to the environments in which indigenous communities live [14]. Local fishing communities have lived and fished in this area for generations and have developed a deep understanding of the natural cycles, patterns, and behaviors of the fish and other aquatic species that they rely on for their livelihoods. Their knowledge includes information on the best times and locations for fishing, the most effective fishing gear and techniques for different species, and the relationships between different fish species and their habitats. Incorporating this community-driven knowledge into research and management strategies could enhance our understanding and facilitate more sustainable management of fish stocks in the MRB. Unfortunately, the perspectives of local communities have often been overlooked or insufficiently acknowledged in both management measures and academic sectors. In recent years, there has been a growing recognition of the value of local ecological knowledge in scientific studies, as well as efforts to integrate this knowledge with scientific knowledge in various parts of the world [15–17].

The role of fish assemblages and biodiversity in maintaining the functionality and resilience of ecosystems in the MRB is crucial, yet their current status remains largely unknown, unmanaged, and unmonitored. Besides, there was no information available in the literature knowledge of local communities regarding the fisheries resources in the MRB. Previous studies on the fish assemblage of the Meghna River have focused only on a small portion near Chandpur district, leaving the rest of the basin understudied [18–21]. To establish a sound management system for commercial fisheries activities, it is crucial to determine the patterns of change in fish abundance and biodiversity across different ecosystems of the MRB, as well as to identify the natural and anthropogenic threats to the basin. Thus, the present study aims to address these gaps by identifying the fish abundance, assemblage, and patterns of shift in fish biodiversity in various ecosystems of the MRB and to identify the associated threats. The findings of this study are expected to provide

essential information to establish an effective management system for the commercial fishery in the MRB and preserve the overall health and resilience of the ecosystem.

## 2. Materials and Methods

A survey was conducted among the fishing communities of the MRB to obtain data for the present study. The study was conducted over a period of six months from August 2015 to January 2016. An overview of the methodology is presented in Figure 1.

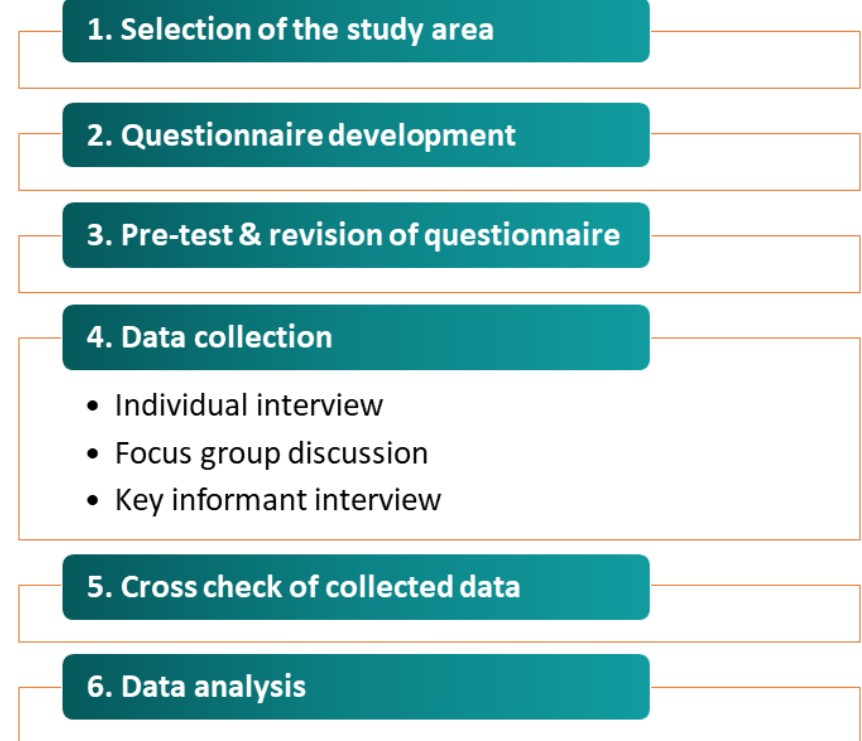

**Figure 1.** Flow diagram showing the methodology used for data collection from fishing communities in the Meghna River Basin.

### 2.1. Description of the Study Sites

The MRB is a large and complex riverine ecosystem that exhibits significant diversity in terms of fish species, fishing practices, and ecological zones. The fishing communities residing along the MRB are also diverse, exhibiting differences in location, fishing practices, and social and cultural norms. To capture the diverse fisheries resources in the region, study sites were carefully selected based on three criteria: the upstream section of the river, the coastal area, and the area with fish sanctuaries. This allowed for a comprehensive understanding of fish assemblage and biodiversity trends across the different ecological zones, including coastal ecosystems, riverine and estuarine ecosystems, and river islands (*Charland*) and their surrounding areas. Based on the abovementioned criteria, participants were chosen from six upazilas or sub-districts (i.e., a local government administrative region that is smaller in size compared to a district) of the MRB, which included Amtali of the Barguna district, Char Fassion and Daulatkhan of the Bhola district, Haimchar of the Chandpur district, Kalapara of the Patuakhali district, and Hijla of the Barisal district (as depicted in Figure 2). These sub-districts are predominantly located along the Meghna River, with the exception of Kalapara and Amtali, which are positioned on the Andharmanik River and Payra River, correspondingly (Table S1).

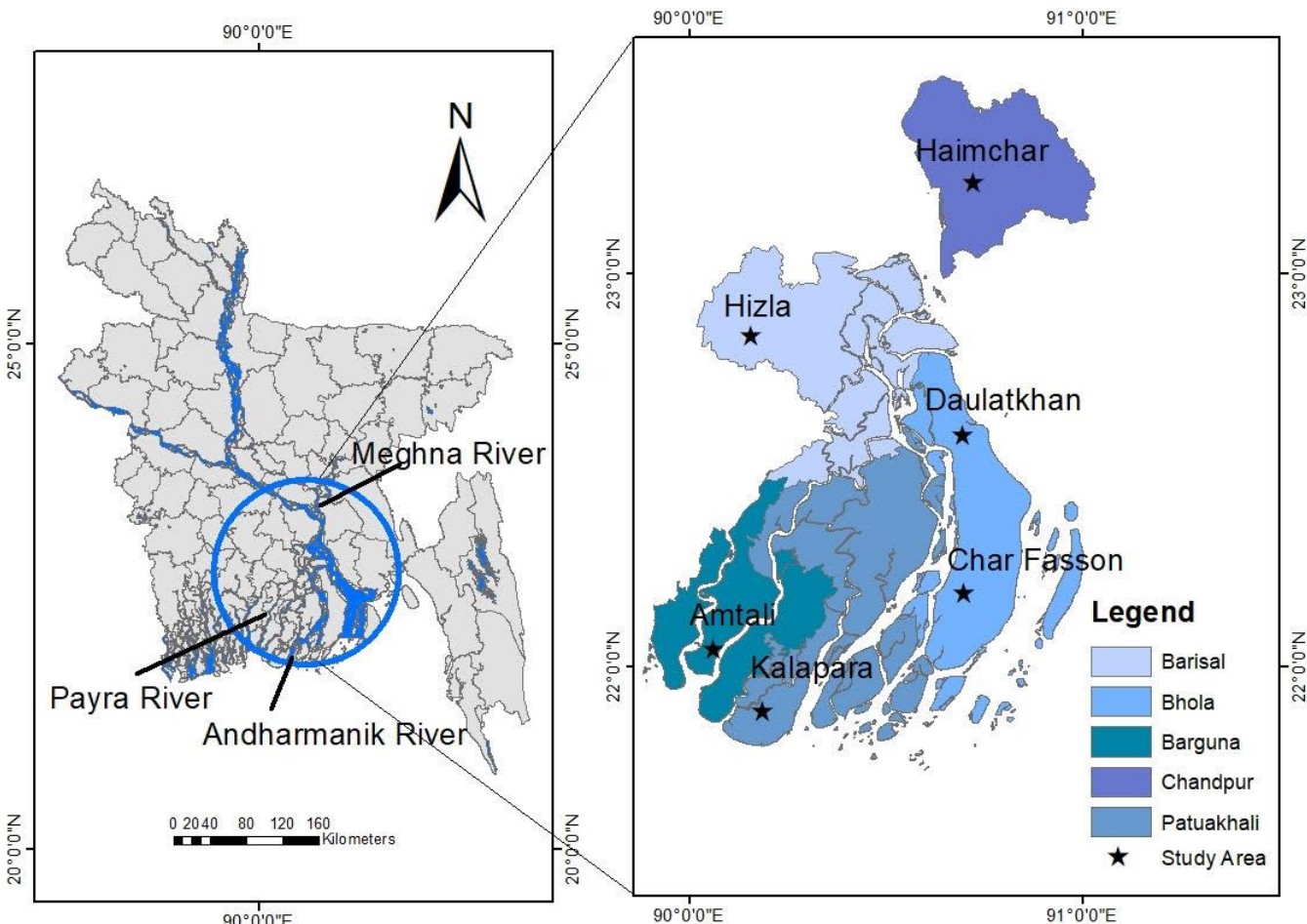

**Figure 2.** Study area and data collection sites in the Meghna River Basin.

## 2.2. Questionnaire Pretesting

To ensure the feasibility and effectiveness of the questionnaire, a preliminary investigation was conducted in two selected areas of the study area. The questionnaire was pretested in the field and subsequently adjusted based on the results.

## 2.3. Sampling and Data Collection Methods

Participatory rural appraisal (PRA) tools such as individual interviews, focus group discussions, and key informant interviews were employed to collect data. These methods provided valuable information and insights from the local fishing communities, which helped to capture a more comprehensive understanding of the ecological and socioeconomic factors that affect the fish assemblage and biodiversity trends in the region. The use of PRA tools also facilitated a participatory and inclusive approach, empowering local stakeholders to be part of the decision-making process for future conservation and management strategies.

For the individual interviews, purposive sampling was used to select respondents from fishing communities of the study area who had a minimum of ten years of fishing experience. Prior to the interviews, the respondents were informed about the purpose and nature of the study, and their consent was obtained. A total of 120 individual interviews with both fishermen and fisherwomen was carried out. Each interview was conducted for a duration of 40–50 min and was carried out in various locations, such as riverbanks, fish markets, and houses. In addition, 25 focus group discussions were conducted with various stakeholder groups, including fishermen and fish traders, to foster interaction and generate new knowledge. Each group consisted of 8–12 people and was supported by a checklist, with each session lasting for 60–80 min. The study also conducted face-to-face

consultations with 36 key informants, which included researchers, government officials, local resource managers, non-governmental organization staff, school teachers, upazila chairman, and local leaders (Table S1). The Focus Group Discussions (FGDs) and Key Informant Interviews (KIIs) were used to cross-check the collected information, ensuring data reliability and consistency.

To supplement primary data, secondary information was collected from various sources, including district fisheries offices, articles, books, journals, and thesis papers. The comprehensive and diverse nature of the data collection process provided a more in-depth and nuanced understanding of the fish assemblage and biodiversity trends in the MRB, helping to develop more informed and effective conservation and management strategies.

*2.4. Information Gathered in the Study*

The information collected included demographic information about the respondents. Respondents were asked to provide information about the present status of highly abundant and commercially exploited fish species, as well as their status 10–20 years earlier. Information was also collected about the fishing gears used by the respondents, including the target species and any negative impacts associated with the use of these gears. Negative impacts of fishing gear have been categorized into three types based on their ecological effects, bycatch rates, and selectivity. These categories are as follows: (a) Less destructive: Minimal ecological impacts, low bycatch rates, and high selectivity; (b) Destructive: Moderate ecological impacts, moderate bycatch rates, and some non-selectivity; and (c) More destructive: Significant ecological impacts, high bycatch rates, and low selectivity. These categorizations were determined by considering expert opinions from fisheries scientists and experienced fishers. To gain a better understanding of the microhabitat in the area, respondents were asked to identify the types of microhabitats present and the abundance of fish in relation to each microhabitat. They were also asked to provide an underlying reason for the abundance of fish in each habitat. Finally, the study sought to identify the drivers that regulate the ecosystems in the MRB, as well as to obtain insights from local communities on how to improve the management of these critical ecosystems (See Supplementary File: Questionnaire S1).

*2.5. Data Analysis*

The collected data were subjected to both quantitative and qualitative analysis. The demographic information was analyzed using descriptive statistics through SPSS (Version 26, IBM, Armonk, NY, USA). The responses collected from the participants were analyzed to determine the frequency and percentage of highly caught species, as well as their temporal and spatial changes, and the influencing factors. Furthermore, a chi-square test for goodness of fit was conducted on the percentage results to identify any significant differences. A significance level of $p < 0.05$ was utilized to determine statistical significance. Graphs were generated with GraphPad Prism (Version 9.0.0). The inductive content analysis method was utilized for analyzing the qualitative data in the study. This method involves identifying patterns, themes, and insights within the data pertaining to seasonal variations, microhabitats, and management strategies. The qualitative data were carefully examined and organized into meaningful categories that reflected different variables. The conservation status of the most caught fish species in the MRB mentioned by the respondents was evaluated using the red list of fish species for Bangladesh, as reported by IUCN Bangladesh [13].

## 3. Results

### 3.1. Demographic Profile of the Respondents

The demographic features of fishers who participated in the study are presented in Table 1. The results reveal that the participants of the study were diverse in terms of demographic characteristics, with a wide range of age, educational, and income levels represented. Among the respondents, a significant portion were in the age group of 26–30 years (31.67%), followed by those in the age group of 31–35 years (26.67%). The majority of participants were male (90.83%) and married (78.33%). The education level of the fishers was generally low, with 40.83% having no education and 52.5% completing only five years of schooling. Fishing was the primary source of income for almost 90% of the households surveyed. Housing conditions varied among participants, with corrugated tin the preferred choice for both wall and roofing construction, while earth or sand the most commonly utilized flooring material. Monthly income ranged from less than BDT 5000 (USD 48) to more than BDT 20,000 (USD 190), with more than half of the participants (53.33%) earning between BDT 5000 and BDT 10,000 (USD 95) per month. The frequency of working days per month varied among participants, with the highest proportion (37.50%) reporting working for 26–30 days per month. Except for age, all other demographic features showed significant differences between categories ($p < 0.05$).

**Table 1.** Demographic profile of the respondents.

| Demographic Characteristics | Frequency | Percentage (%) | $\chi^2$ | *p* Value |
|---|---|---|---|---|
| **Age (years)** | | | | |
| <25 | 21 | 17.50 | 5.000 | 0.172 |
| 26–30 | 38 | 31.67 | | |
| 31–35 | 32 | 26.67 | | |
| >36 | 29 | 24.17 | | |
| **Gender** | | | | |
| Female | 11 | 9.17 | 80.033 | 0.000 |
| Male | 109 | 90.83 | | |
| **Marital status** | | | | |
| Married | 94 | 78.33 | 110.150 | 0.000 |
| Single | 17 | 14.17 | | |
| Divorce | 9 | 7.5 | | |
| **Educational status** | | | | |
| No education (illiterate and can sign only) | 49 | 40.83 | 94.000 | 0.000 |
| Five years of schooling | 63 | 52.50 | | |
| Eight years of schooling | 7 | 5.83 | | |
| Ten years of schooling | 1 | 0.83 | | |
| **Housing conditions (n = 120)** | | | | |
| **House building materials** | | | | |
| **Wall materials** | | | | |
| Cane/palm/trunks | 7 | 5.83 | 342.667 | 0.000 |
| Corrugated tin | 105 | 87.50 | | |
| Cement and bricks | 1 | 0.83 | | |
| Wood planks and shingles | 5 | 4.17 | | |
| Others | 2 | 1.67 | | |
| **Roofing materials** | | | | |
| Thatch palm leaf | 7 | 5.83 | 368.417 | 0.000 |
| Bamboo with mud | 2 | 1.67 | | |
| Hardboard/polythene | 2 | 1.67 | | |
| Corrugated tin | 108 | 90.00 | | |
| Roofing shingles | 1 | 0.83 | | |

**Table 1.** *Cont.*

| Demographic Characteristics | Frequency | Percentage (%) | $\chi^2$ | *p* Value |
|---|---|---|---|---|
| **Flooring materials** | | | | |
| Earth/sand | 111 | 92.50 | 189.650 | 0.000 |
| Wood planks | 8 | 6.67 | | |
| Palm/bamboo | 1 | 0.83 | | |
| **Monthly income (BDT)** | | | | |
| ≤5000 (USD * 48) | 25 | 20.83 | 60.733 | 0.000 |
| 5000–10,000 (USD 48–95) | 64 | 53.33 | | |
| 10,000–20,000 (USD 95–190) | 26 | 21.67 | | |
| ≥20,000 (USD 190) | 5 | 4.17 | | |
| **Working days per month** | | | | |
| ≤15 | 24 | 20.00 | 12.867 | 0.005 |
| 16–20 | 32 | 26.67 | | |
| 21–25 | 19 | 15.83 | | |
| 26–30 | 45 | 37.50 | | |

* USD: United States Dollar (1 USD = 105 BDT).

### 3.2. Ecological Diversity of Fishing Grounds in the Study Area

The fishermen in the study areas catch fish from five different types of ecosystems. The majority of respondents catch fish from the riverine ecosystem (60.83%), followed by the charland (47.50%), the Bay of Bengal (11.67%), the estuarine (8.33%), and the mangrove ecosystems (4.17%), as depicted in Figure 3.

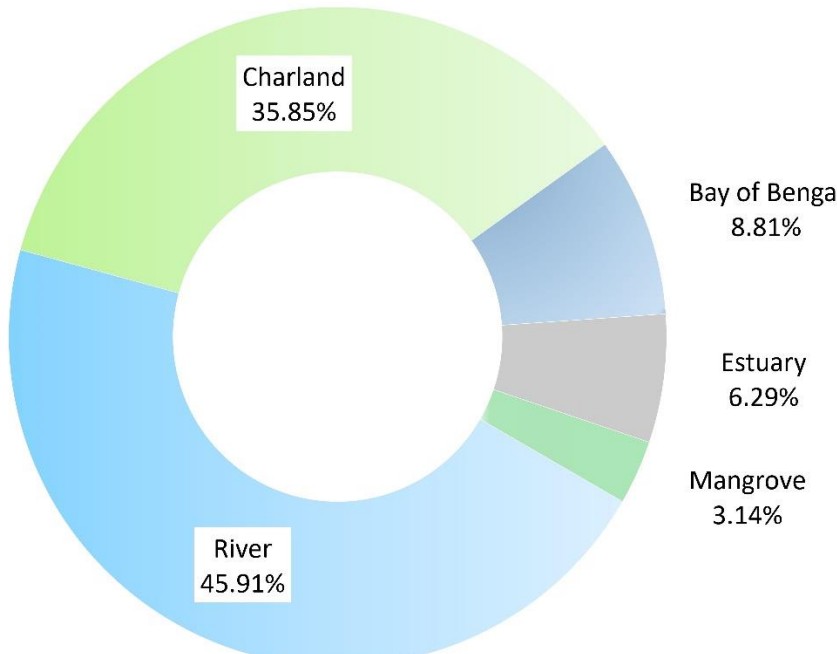

**Figure 3.** Distribution of fishing grounds where fishers of the study area harvest fish (multiple answers).

### 3.3. Highly Captured Fish Species in the MRB

During the survey, participants were asked to identify the ten most commonly harvested fish species in the MRB. The results suggest that Hilsa shad (*Tenualosa ilisha*; Hamilton, 1822) and Croakers pama (*Otolithoides pama*; Hamilton, 1822) are the most frequently caught fish species in the MRB, with 95% and 88% of respondents reporting catching them, respectively. Lanceolate goby (*Pseudapocryptes elongatus*; Cuvier, 1816) and Paradise threadfin (*Polynemus paradiseus*; Linnaeus, 1758) are also significant species, with 61% and 57% of respondents reporting capturing these species, respectively. Meanwhile, the other species identified, including Tank goby (*Glossogobius giuris*; Hamilton,

1822), Greenback mullet (*Planiliza subviridis*; Valenciennes, 1836), Gangetic hairfin anchovy (*Setipinna phasa*; Hamilton, 1822), Ganges River sprat (*Corica soborna*; Hamilton, 1822), Flathead sillago (*Silloginopsis panijus*; Hamilton, 1822), and Neglected grenadier anchovy (*Coilia neglecta*; Whitehead, 1968) were cited by a smaller proportion of respondents (Figure 4). In addition, a chi-square test indicates a significant variation in the percentages of the top ten caught fish species ($\chi^2$ = 234.056, $p$ < 0.001). This suggests that there are substantial differences in the relative abundance or catch rates of these fish species as reported by the fishers.

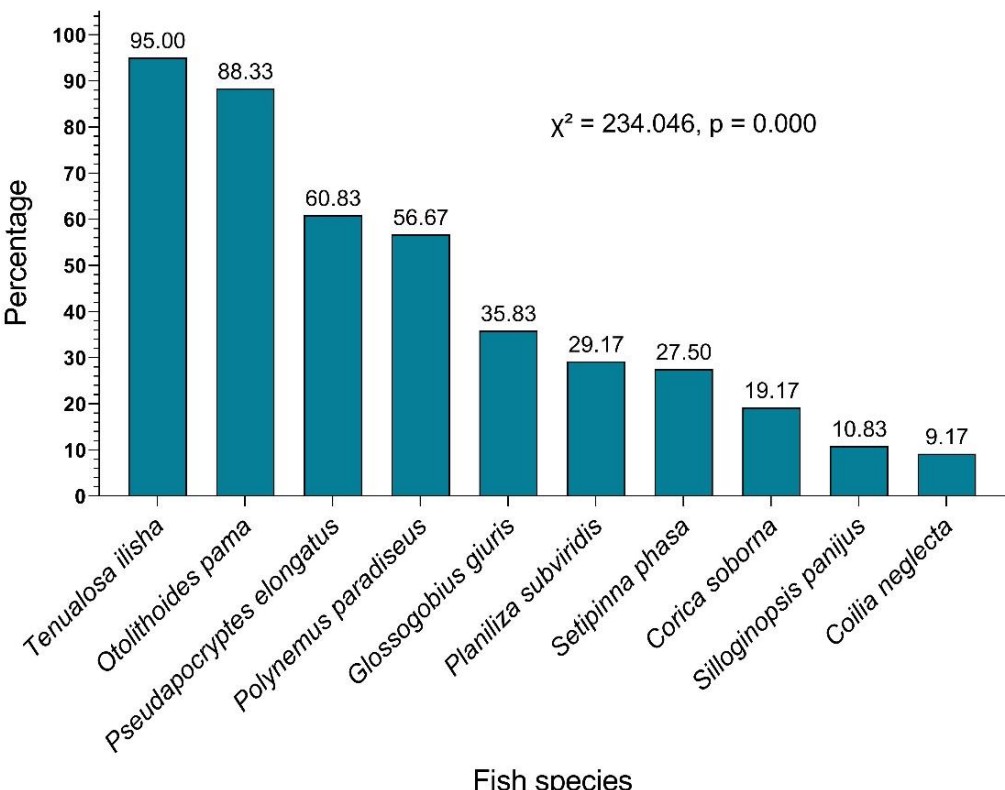

**Figure 4.** Ten most frequently captured fish species in the Meghna River Basin.

### 3.4. Spatial Variation of Frequently Caught Fish Species in the MRB

Table 2 presents the variation in the most frequently caught fish species in the MRB across six different areas, as reported by the survey respondents. The results show that there are spatial differences in the catch rates of these fish species within the MRB. Hilsa shad (*T. ilisha*) and Croakers pama (*O. pama*) are the most frequently caught fish species in all areas surveyed, with a range of 70–100% of respondents citing catching these two species. The chi-square test implies that the availability of these two species does not significantly vary across the different areas. Lanceolate goby (*P. elongatus*) and Paradise threadfin (*P. paradiseus*) are also important species, with high percentages of respondents reporting catching them in most areas. However, there is significant variation in the catch rates of other species across different areas. For example, the Tank goby (*G. giuris*) is most frequently caught in Hizla (85%) and Haimchar (80%), and the Greenback mullet (*P. subviridis*) is most frequently caught in Daulatkhan (95%).

**Table 2.** Spatial variation of commonly caught fish species in the Meghna River Basin.

| Fish Species | Percentage of Respondents | | | | | | $\chi^2$ | *p* Value |
| | Amtali | Char Fasson | Daulatkhan | Haimchar | Kalapara | Hizla | | |
|---|---|---|---|---|---|---|---|---|
| *Tenualosa ilisha* | 95 | 95 | 100 | 100 | 80 | 100 | 3.158 | 0.676 |
| *Otolithoides pama* | 95 | 95 | 100 | 80 | 70 | 90 | 7.170 | 0.208 |
| *Pseudapocryptes elongatus* | 15 | 35 | 75 | 90 | 60 | 90 | 76.781 | 0.000 |
| *Polynemus paradiseus* | 70 | 75 | 95 | 0 | 20 | 80 | 48.164 | 0.000 |
| *Glossogobius giuris* | 0 | 15 | 10 | 80 | 25 | 85 | 123.953 | 0.000 |
| *Planiliza subviridis* | 0 | 0 | 95 | 65 | 0 | 15 | 56.000 | 0.000 |
| *Setipinna phasa* | 15 | 55 | 70 | 0 | 5 | 20 | 94.848 | 0.000 |
| *Corica soborna* | 0 | 0 | 0 | 60 | 40 | 15 | 26.522 | 0.000 |
| *Silloginopsis panijus* | 0 | 0 | 55 | 0 | 10 | 0 | 31.154 | 0.000 |
| *Coilia neglecta* | 0 | 10 | 5 | 10 | 25 | 5 | 24.545 | 0.000 |

### 3.5. Temporal Variation in Frequently Caught Fish Species in the MRB

The survey included a question asking fishers to identify the ten fish species that were most frequently caught in the MRB 10 to 20 years ago. Among these species, Giant perch (*Lates calcarifer*; Bloch, 1790) was the most commonly caught fish, reported by 79% of the respondents. Wallago (*Wallago attu*; Bloch & Schneider, 1801) and Long whiskers catfish (*Sperata aor*; Hamilton, 1822) were the second and third most frequently caught fish species, with percentages of 73% and 68%, respectively. Yellowtail catfish (*Pangasius pangasius*; Hamilton, 1822), Rita (*Rita rita*; Hamilton, 1822), Clown knifefish (*Chitala chitala*; Hamilton, 1822), Corsula (*Rhinomugil corsula*; Hamilton, 1822), Flathead grey mullet (*Mugil cephalus*; Linnaeus, 1758), Chinese silver pomfret (*Pampus chinensis*; Euphrasen, 1788), and Surf bream (*Acanthopagrus latus*; Houttuyn, 1782) were the other species cited by the respondents (Figure 5). The chi-square test denotes significant temporal variation in the availability of these fish species ($\chi^2$ = 98.546, *p* < 0.001).

### 3.6. Seasonality and Conservation Status of Commonly Caught Fish Species

The conservation status of the fish species identified as the most commonly caught was assessed using the International Union for Conservation of Nature (IUCN) Red List (Table 3). In Bangladesh, for the present-day fish species, it is observed that nine out of ten species fall into the least concern (LC) category. Six of the ten species are also classified as LC globally, while the status of the remaining four species is listed as not evaluated (NE). On the other hand, among the ten species listed as frequently caught species before 10–20 years, six are currently classified as threatened or near threatened in Bangladesh. Globally, most of these species are classified as LC or NE, with only two falling under the near threatened (NT) category, namely Wallago and Clown knifefish.

The study participants were tasked with identifying the duration of availability of the most commonly caught fish species in the MRB (Table 3). The results showed that these species were available for varying lengths of time, ranging from three to six months in a year. Most of the species were reported to be available during the winter season, with the exception of the Hilsa shad, which was available during the monsoon season spanning from June to October.

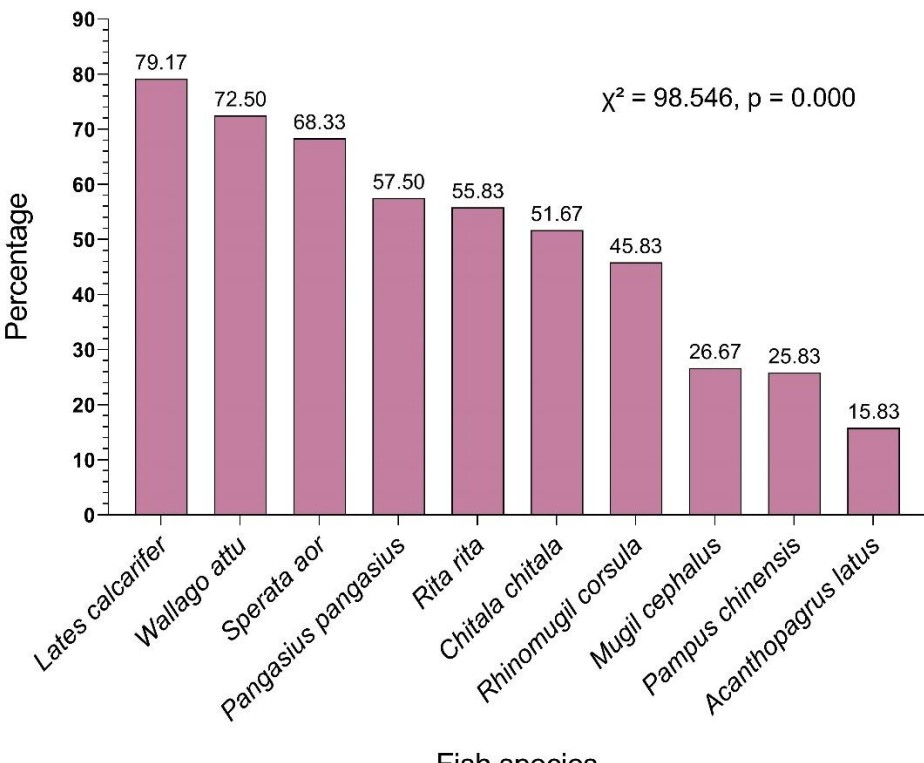

**Figure 5.** Top ten fish species frequently caught in the Meghna River Basin 10–20 years ago, as reported by fishers.

**Table 3.** Conservation and availability status of top ten fish species in the Meghna River Basin [13].

| Scientific Name | Local Name | English Name | IUCN Status * | | Duration of Availability |
| --- | --- | --- | --- | --- | --- |
| | | | Global | BD | |
| **Currently most caught fish species** | | | | | |
| *Tenualosa ilisha* | Ilish | Hilsa shad | LC | LC | June to October |
| *Otolithoides pama* | Poa/Pama | Croakers pama | NE | LC | January to March |
| *Pseudapocryptes elongatus* | Chewa | Lanceolate goby | NE | LC | December to March |
| *Polynemus paradiseus* | Taposhi | Paradise threadfin | NE | LC | December to March |
| *Glossogobius giuris* | Bele/Baila | Tank goby | LC | LC | January to March |
| *Planilizasubviridis* | Bata | Greenback mullet | LC | LC | November to March |
| *Setipinna phasa* | Phasa | Gangetic hairfin anchovy | LC | LC | October to March |
| *Corica soborna* | Kachki | Ganges river sprat | LC | LC | December to February |
| *Sillaginopsis panijus* | Tulardandi | Flathead sillago | NE | NT | November to January |
| *Coilia neglecta* | Olua | Neglected grenadier anchovy | LC | LC | December to February |
| **Previously most caught fish species** | | | | | |
| *Lates calcarifer* | Koral | Giant perch | NE | NE | - |
| *Wallago attu* | Boal | Wallago | NT | VU | - |
| *Sperata aor* | Ayre | Long whiskers catfish | LC | VU | - |
| *Pangasius pangasius* | Pangus | Yellowtail catfish | LC | EN | - |
| *Rita rita* | Rita | Rita | LC | EN | - |
| *Chitala chitala* | Chital | Clown knifefish | NT | EN | - |
| *Rhinomugil corsula* | Khorshula | Corsula | LC | LC | - |
| *Mugil cephalus* | Bhangon | Flathead grey mullet | LC | LC | - |
| *Pampus chinensis* | Rupchanda | Chinese silver pomfret | NE | NT | - |
| *Acanthopagrus latus* | Datina | Surf bream | NE | DD | - |

* LC: least concern; NE: not evaluated; NT: near threatened; VU: vulnerable; EN: endangered; DD: data deficient; BD: Bangladesh.

### 3.7. Relationship between Microhabitats and Fish Abundance in the MRB

Microhabitats are small-scale habitats within the larger river ecosystem that are characterized by distinct physical and environmental conditions. The participants of the study have recognized several microhabitats in the MRB, including stiff slope, gentle slope, scour, river trench, inundated river island, and erosion of dike and their ecological significance (Figure 6).

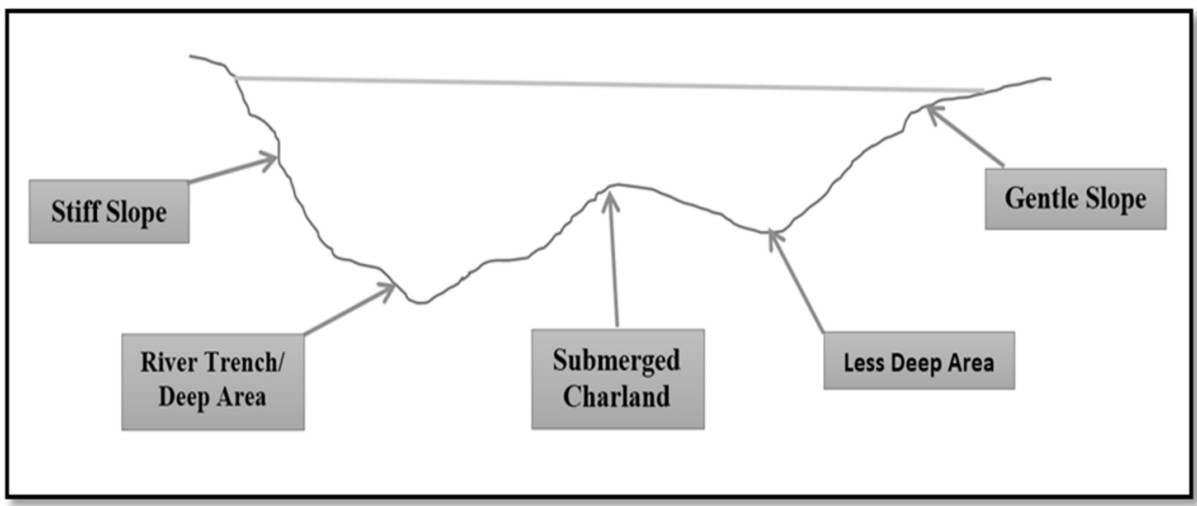

**Figure 6.** Microhabitats in the Meghna River Basin identified by the respondents.

These microhabitats within the MRB display a diverse array of characteristics, encompassing variations in water depths, including both shallow and deep areas. Moreover, some of these microhabitats undergo seasonal fluctuations in water levels. Fish populations utilize these microhabitats for a range of essential activities, including feeding, breeding, nursing, and migration pathways. For instance, the stiff slopes found in the basin provide an optimal feeding environment for species such as Hilsa shad (*T. ilisha*) and Long whiskers catfish (*S. aor*). On the other hand, gentle slopes function as important grazing and nursing grounds for juvenile fish of various species. Additionally, inundated river islands play a crucial role in supporting the nursing, grazing, and breeding activities of young fish species, including Tank goby (*G. giuris*) and Greenback mullet (*P. subviridis*) (Table 4).

**Table 4.** Microhabitats and their significance for fish species in the Meghna River Basin reported by respondents.

| Microhabitats | Characteristics | Available Fishes | Significance |
|---|---|---|---|
| Stiff slope | Stiff, eroded, high velocity, turbulent, and turbid water | *Tenualosa ilisha, Otolithoides pama, Pangasius pangasius, Wallago attu, Polynemus paradiseus, Sperata aor,* and *Glossogobius giuris* | Predation and feeding |
| Gentle slope | Shallow water, low velocity, less turbid, and highly productive | Juveniles of *Tenualosa ilisha, Sperata aor* and *Pangasius pangasius* | Grazing and nursing |
| Scour | Deep water, cool, and turbulent | *Pangasius pangasius, Sperata aor* and adult *Tenualosa ilisha* | Feeding and shelter |
| River trench | Deep water and high velocity | *Tenualosa ilisha, Otolithoides pama, Polynemus paradiseus, Pangasius pangasius* and *Sperata aor* | Shelter and resting |
| Inundated river island | Shallow, seasonal inundation | Young *Tenualosa ilisha, Otolithoides pama, Polynemus paradiseus, Pseudapocryptes elongatus, Glossogobius giuris, Setipinna phasa, Corica Soborna* and *Planiliza subviridis* | Nursing, grazing, and breeding |
| Erosion of dike | Shallow water seasonal connectivity | *Glossogobius giuris, Planiliza subviridis* and *Sperata aor* | Seasonal fish passage |

### 3.8. Fishing Gears Used in the Meghna River Basin

The results of this study demonstrate that the fishing communities of the MRB employ a diverse array of fishing gear (Table 5). These gears can be broadly categorized into three main groups: fish nets, traps, and hooks and lines. Among these categories, fish nets were found to be the most commonly used fishing gear by the fishermen in the region, catching species such as Hilsa shad, Croakers pama, Yellowtail catfish, and Giant perch. Gillnets and seine nets are the predominant types of fish nets used in the MRB, and have various subvarieties, each targeting specific species. These nets require significant investments and operating costs, and are therefore primarily used by fishermen with better economic conditions. In contrast, lower-cost fishing gears, such as drag nets, cast nets, and traps, tend to be used more by relatively impoverished fishermen. Additionally, the study evaluated the ecological impacts of these fishing gears and found that one type of gillnet, known locally as *current jal*, and two types of seine nets, namely *ber jal* and *jogot ber jal*, had higher negative impacts due to their small mesh size, which results in the capture of various fish species, including juveniles.

**Table 5.** Fishing gears used in the Meghna River Basin with their target species and associated negative impacts.

| Gears | | | Major Species Caught | Negative Impacts * |
|---|---|---|---|---|
| Category | English Name | Local Name | | |
| Fish nets | Gill net | Chandi/Sine jal | Mainly Hilsa shad, and other fishes include Croakers pama, Yellowtail catfish, Giant perch | + |
| | | Poa jal | Mainly Croakers pama, and others include Hilsa shad, Tank goby, Yellowtail catfish, Paradise threadfin, Greenback mullet, Long whiskers catfish, Rita, Lanceolate goby | + |
| | | Koral/Fash jal | Mainly Giant perch, and others include Yellowtail catfish, Long whiskers catfish, Rita, Wallago, Surf bream | + |
| | | Current jal | Hilsa shad, Croakers pama, Tank goby, Yellowtail catfish, Paradise threadfin, Greenback mullet, Long whiskers catfish, Rita, Wallago, Long whiskers catfish | ++ |
| | Seine net | Ber Jal | Hilsa shad, Prawns, Lanceolate goby, Greenback mullet, Tank goby, Flathead sillago, Ganges River sprat, and other small fishes | +++ |
| | | Gulti/Jagat ber Jal | Hilsa shad, Wallago, Croakers pama, Tank goby, Long whiskers catfish, Rita, Yellowtail catfish, and other small fishes | ++ |
| | | Kona Jal | Hilsa shad, Croakers pama, Long whiskers catfish, Yellowtail catfish, and Giant perch | + |
| | Set bag net | Behundi jal | Hilsa shad, Prawns, Croakers pama, Lanceolate goby, Tank goby, and other small fishes | +++ |
| | Drag net | Chewa jal | Lanceolate goby, Prawns, and Tank goby | + |
| | | Moia jal | Prawns, Croakers pama, Lanceolate goby, Tank goby, Yellowtail catfish | ++ |
| | Cast net | Jhaki jal | Prawns, Tank goby, Croakers pama, Greenback mullet | + |
| Traps | Pot | Pangus chai | Yellowtail catfish | + |
| | | Ichar chai | Prawns | + |
| Hook & line | Longline | Borshi | Croakers pama, Paradise threadfin, Long whiskers catfish, Rita, Yellowtail catfish, Wallago | ++ |

* '+'—less destructive; '++'—destructive; '+++'—more destructive.

### 3.9. Drivers of Negative Impacts on Meghna River Basin Ecosystems

To better understand the factors contributing to harmful impacts on the aquatic resources and ecosystems of the MRB, respondents were asked to identify these drivers. The responses revealed a diverse range of factors affecting the region, which were subsequently organized into distinct categories for analysis (Figure 7). The two most cited drivers were overexploitation (65%) and non-compliance of fishing laws (62%), highlighting the significant role that human activities play in the degradation of the region's ecosystems. One fisherman from Haimchar spoke about the challenge, saying "We know that the fish stocks are declining, but we have no other livelihood option but to continue fishing because we need to feed our families". Another key informant pointed out, "Overfishing has been a common practice in this region for years, which has led to the decline of fish stocks and harmed the aquatic ecosystem. Despite the existence of fishing laws, non-compliance

with these laws is rampant and often goes unchecked, exacerbating the problem". Other important drivers mentioned by respondents include the lack of alternative jobs (44%), construction of dams and embankments (38%), riverbank erosion (28%), navigation and transportation (27%), siltation (23%), and reduced river flow (22%). Ecotourism, pollution, climate change, and high salinity were also identified as contributing factors. These results indicate the complex and multifaceted nature of the challenges facing the MRB and the need for comprehensive and integrated management strategies to address these issues.

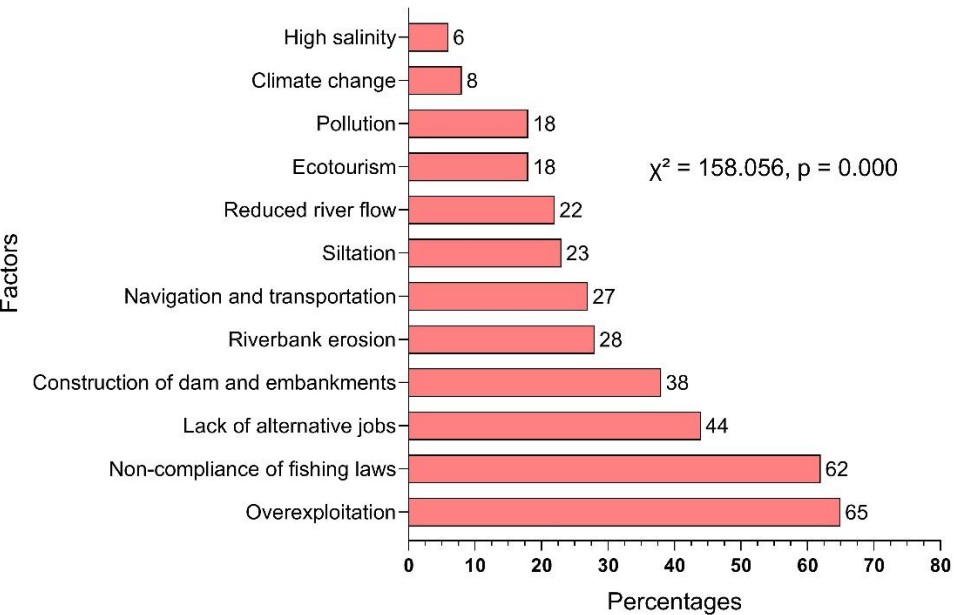

**Figure 7.** Factors affecting ecosystems and aquatic resources in the Meghna River Basin identified by respondents.

### 3.10. Management Measures Suggested by Fishermen of the Meghna River Basin

The study aimed to elicit the opinions of local fishermen on measures that could be taken to conserve the aquatic resources and promote sustainable utilization of the MRB. Respondents provided a range of suggestions, including the need to stop using illegal fishing gears like gill nets and estuarine set bag nets, as highlighted by a fisherman from Kalapara. "It is essential to stop using illegal gear such as gill nets and estuarine set bag nets which have harmful impacts on fish populations", he stressed. Another fisherman from Hizla emphasized the importance of government support during the fishing ban period to reduce fishing pressure and allow fish stocks to replenish. Other measures suggested by the respondents included the construction of sluice gates, dredging of the river to increase depth and current, and arranging training for fishermen on fishing techniques, fish culture, and other technical skills. Respondents also emphasized the importance of creating alternative income-generating opportunities during the ban period, reducing fishing pressure by establishing sanctuaries for conserving juvenile fishes, proper enforcement of laws, and demarcation of areas for fishing. Additionally, they suggested appropriate mesh sizes for fishing gears, licensing of fishing boats/crafts, and reducing industrial fishing with mechanized boats. The importance of reducing fishing pressure and ceasing industrial fishing with mechanized boats to ensure sustainable fishing for future generations was emphasized by a knowledgeable fisherman from Char Fasson. These suggestions underscore the potential for a range of management strategies to promote sustainable fishing practices and conservation of the aquatic resources of the study area.

### 4. Discussion

This study aimed to investigate the local ecological knowledge of fishing communities in the MRB in Bangladesh. Fishermen in the MRB, like fishermen from other regions, have

a longstanding tradition of engaging in discussions about fish ecology and behavior. These conversations and anecdotes shared among the fishermen can offer valuable insights into various aspects of fisheries and fish resources. Although scientists have often regarded such information as anecdotal and less scientifically rigorous, there is a growing recognition of its potential as a valuable complementary source of information [22]. Fishermen's tales and observations can yield important information on the size and abundance of fish caught, fish behavior, historical fisheries trends, and the overall status of fish resources. This type of information may extend beyond the scope of conventional scientific data collected through standardized approaches. It can offer unique perspectives on past abundance patterns of target fish, providing historical insights that predate the scientific recording of data. Incorporating the knowledge and observations shared by fishermen into scientific research and management practices can lead to a more comprehensive understanding of the fisheries ecosystem [15]. By acknowledging the value of fishermen's knowledge and experiences, and fostering collaborative partnerships between fishermen and scientists, a more holistic approach to fisheries management could be established. This approach ensures the sustainable utilization and conservation of fish resources in the MRB and beyond.

The fishing communities in the MRB are facing challenging socioeconomic conditions, including a low level of literacy, limited access to basic amenities, and a higher dependency on fisheries resources. Additionally, the income of these communities is marked by volatility and uncertainty, which only exacerbates their economic instability. These findings are in line with previous research conducted across many areas of Bangladesh [23–27], underscoring the necessity for greater attention to the social aspects of fishing communities and their impact on the successful management of fisheries resources and the well-being of these communities.

This study reveals that the Hilsa shad (*T. ilisha*) is the most frequently caught fish species in the MRB, highlighting its significance in the local economy and culture. This is consistent with the fact that Hilsa shad is the largest single-species capture fishery in Bangladesh, accounting for almost 13% of the country's total fish production [2]. In addition, this fishery supports a large number of individuals, with 0.5 million fishers directly employed and another 2.5 million individuals associated with its value chain [28]. Other highly captured fish species in the MRB include Croakers pama (*O. pama*), Lanceolate goby (*P. elongatus*), Paradise threadfin (*P. paradiseus*), Tank goby (*G. giuris*), Greenback mullet (*P. subviridis*), Gangetic hairfin anchovy (*S. phasa*), Ganges River sprat (*C. soborna*), Flathead sillago (*S. panijus*), and Neglected grenadier anchovy (*C. neglecta*). Three different studies conducted on three rivers of the MRB, namely the Meghna River [20], the Andharmanik River [29] and the Payra River [30], have reported the presence of these species, although their catch composition varied slightly from that reported by fishers in the present study. This variation in dominant fish species can be attributed to several factors, such as variations in fishing practices, gear types, fishing locations, environmental conditions, and time of day and season when fishing activities are conducted.

There has been a significant change in the most captured fish species composition in the MRB over the past decade. Six of the ten species previously listed as frequently caught in the MRB are now classified as threatened or near threatened in Bangladesh. These findings are in line with previous studies. For example, approximately 20% of the 107 fish species in the Meghna River [21], 21% of 81 fish species in the Andharmanik River [29], and nearly 17% of 61 fish species in the Payra River [30] are recorded as threatened. The observed changes in the composition of the most frequently caught fish species in the MRB may have significant implications for the fishing industry and ecosystem dynamics in the region, emphasizing the urgent need to protect and conserve these threatened species and promote sustainable fishing practices.

In the early 2000s, Hilsa shad production in the country had sharply declined, and the species was no longer among the most frequently caught fish in the MRB [2]. To address this issue, the Government of Bangladesh initiated the Hilsa Fishery Management Action Plan

(HFMAP) in 2005, focusing on several key areas. These included establishing sanctuaries to protect juvenile brood fish during the peak season, eradicating harmful fishing gears, protecting migratory routes, controlling overfishing, providing food incentives during ban periods, and promoting alternative income-generating activities for fishers [31,32]. The implementation of the HFMAP has been successful in reviving the Hilsa shad fishery in the MRB. However, the lack of similar initiatives for other threatened fish species in the region is a cause for concern. It is imperative that such initiatives be implemented as part of a broader, integrated management approach that accounts for both ecological and socio-economic factors.

Spatial variations in the most frequently caught fish species in the MRB can be attributed to differences in fishing practices, local ecological conditions, and species availability [21,33]. This information can be used by policymakers and fishery managers to develop tailored management strategies that promote sustainable fishing practices and protect the region's most important fish species. Moreover, the study also highlights the importance of considering the duration of fish species availability for sustainable fisheries management. The results indicate that the most caught fish species in the MRB are available for three to six months per year, with Hilsa shad having the longest availability of five months during the monsoon season.

The distribution and abundance of fish species in relation to microhabitats within the MRB reported by fishers have significant implications for the conservation and management of the river ecosystem. Microhabitat influences fish abundance by providing suitable conditions for feeding, sheltering, and reproduction. Microhabitat selectivity by fish species is an important factor that underpins regional indicators of fish abundance [34]. The identification of microhabitats and their ecological significance can inform effective river ecosystem management and conservation efforts. In particular, it can aid in the development of targeted conservation strategies that focus on critical microhabitats for the fish species. Furthermore, this information can be used to guide sustainable fishing practices that prioritize specific microhabitats at appropriate times of the year, reducing the impact of fishing on fish populations in the MRB [35].

Fishermen in the MRB have access to diverse fishing grounds encompassing both freshwater and saltwater ecosystems, and employ a range of fishing gears, many of which are traditional low-tech gears. The use of these fishing gears is a common practice among fishers in Bangladesh, as reported in several studies [36–38]. Fishermen utilize these fishing gears because they are cost-effective, readily accessible, and have proven to be efficient in capturing their desired target species. However, the ecological impacts of these fishing gears cannot be ignored, as they can have negative effects on fish populations and the overall health of the riverine ecosystem. The study found that certain fishing gears, such as gillnets and seine nets, have higher negative impacts due to their small mesh size [39]. These gears can capture not only the target species but also various other fish species, including juveniles, leading to a decline in their populations. This result emphasizes the necessity for efficient management and conservation strategies to regulate the use of destructive fishing gears and safeguard fish populations in the MRB.

This study provides valuable insights into the complex and diverse drivers of ecosystem degradation in the MRB. The findings reveal that local fishers in the region perceive a range of factors as contributing to negative impacts on the ecosystem, with overexploitation and non-compliance with fishing laws being the most commonly cited. These drivers have been identified as significant contributors to ecosystem degradation in aquatic systems worldwide, as noted in previous research [40–42]. Additionally, the effects of these drivers are amplified by various other challenges, such as the absence of alternative employment, the building of dams, the erosion of riverbanks, navigation, sediment accumulation, decreased river flow, ecotourism, pollution, and climate change [43].

Enforcing fishery regulations is crucial for the sustainable management of aquatic resources in the MRB. Despite the existence of fishing laws, such as the Protection and Conservation of Fish Act, 1950, the Protection and Conservation of Fish Rules, 1985, and

the Marine Fisheries Act, 2020, non-compliance with these laws is rampant and often goes unchecked, leading to further degradation of the ecosystem [44]. The government must take urgent action to ensure compliance with current regulations and to develop and implement new conservation measures to protect and sustain the region's fisheries resources. Collaborative efforts involving the fishing communities, government agencies, and non-governmental organizations can help to promote sustainable fishing practices and conservation of the MRB's fishery resources. Such efforts can contribute not only to the ecological well-being of the region but also to the economic and social well-being of the local fishing communities.

It is worth noting that the present research has limitations that should be acknowledged. The study relied solely on self-reported data from fishing communities, which may have introduced biases or inaccuracies in the accumulated results. Additionally, the study did not examine the ecological impacts of specific fishing practices or gears, which could be a subject for future research. Future studies could also explore the perspectives of other stakeholders, such as government agencies, non-governmental organizations, and consumers, to gain a more comprehensive understanding of the complex issues involved in the sustainable management of aquatic resources in the MRB. Moreover, it is important to acknowledge that the lack of statistical analysis performed on the accumulated data limits the depth of quantitative interpretation of the findings of the study. Nonetheless, the study provides critical insights into the perception of local fishing communities regarding the fisheries resources of the MRB and can serve as a foundation for future research and conservation efforts.

## 5. Conclusions

The present study offers important insights into the knowledge of local fishers about the fisheries resources in the MRB. The study provides a comprehensive account of the highly captured fish species and their spatial and temporal variations through the perspectives of fishing communities. It is alarming to note that among the ten fish species that were frequently caught in the past 10–20 years, six of them are currently classified as threatened or near threatened in Bangladesh. The study emphasizes the significance of microhabitats in river ecosystem management and conservation efforts. Furthermore, it has been reported that lower-cost fishing gears, which are associated with various harmful impacts, are predominantly used by fishers. Overfishing and non-compliance with fishing laws were identified as significant factors negatively affecting the ecosystems in the study area. As such, the findings highlight the need for immediate enforcement of existing fishery regulations and the implementation of new conservation measures to ensure the sustainable management of aquatic resources in the region. The results of the study render a solid foundation for future research and conservation efforts aimed at promoting ecological, economic, and social well-being in the region.

**Supplementary Materials:** The following supporting information can be downloaded at: https://www.mdpi.com/article/10.3390/su151411466/s1, Table S1. Summary of study areas and methodological tools utilized in the study; Questionnaire S1: Questionnaire for Fishers.

**Author Contributions:** Conceptualization, M.A.S., M.N., B.K.B., M.A.W. and M.K.; formal analysis, M.A.S. and M.A.H.; funding acquisition, P.S., M.A.W. and M.K.; methodology, M.A.S. and B.K.B.; software, M.A.H.; supervision, B.K.B. and M.K.; writing—original draft preparation, M.A.S. and M.A.H.; writing—review and editing, P.S., M.N. and M.M.H.M. All authors have read and agreed to the published version of the manuscript.

**Funding:** The funding for this study was provided by the WorldFish Bangladesh through the ECOFISH[BD] project.

**Institutional Review Board Statement:** Not applicable.

**Informed Consent Statement:** Informed consent was obtained from all subjects involved in the study.

**Data Availability Statement:** The corresponding author can provide the data accumulated for the current study upon request.

**Acknowledgments:** The authors would like to acknowledge the invaluable contributions of the ECOFISH^BD team members during the data collection and analysis. We would also like to express our gratitude to the study participants for their time and willingness to share their knowledge and experiences with us. This research would not have been possible without their invaluable contributions.

**Conflicts of Interest:** The authors disclose that the research was conducted without any existing or potential conflicts of interest, including commercial or financial relationships that could influence the objectivity or integrity of the study.

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
