# Peer review of "Community-Driven Insights into Fish Assemblage, Microhabitats, and Management Strategies in the Meghna River Basin of Bangladesh"

_sustainability, doi:10.3390/su151411466_

Round 1

Reviewer 1 Report

Dear Authors,

The study highlights the importance of microhabitats in the management and conservation of river ecosystems and emphasizes the need to enforce existing fishery regulations and implement new conservation measures for sustainable management of aquatic resources in the region (MRB).

However, I think some minor changes and corrections are necessary in the article. First of all, the names of the fish species should be written next to who classified them, or at least be expressed next to the name of the species when mentioned for the first time and there is no need to write it again the second time. For example, Planiliza subviridis (Valenciennes, 1836)

Secondly, the species Liza subviridis has been accepted as "Planiliza subviridis", it should be changed in Figure 4, Figure 5, Table 3, and in the text.

The species Notopterus chitala has been accepted as Chitala chitala, it should be changed and fixed in Figure 6 and Table 3.

In Figure 5, "Pseudapocryptes elongates" is mistakenly written, it should be corrected as "Pseudapocryptes elongatus".

In Table 3, it should be written as "Corico soborna", a capital "S" has been used incorrectly, a lowercase "s" should be used instead.

Best Regards.

Reviewer 2 Report

The manuscript (MS) had a variety of well-diversified data matrix based on questionnaire on the fisheries for aspects of socio-economic fisheries of the different communities. 

The presentation and analyses of the data is not good enough to makes wide sound to interest. Some points arisen as comments below:

Main problem of the MS is  that the data were however not subjected to statistical non-parametric uni- or multivariate analyses.  ANOVA, or MANOVA for difference, GAM for the effective. 

Introduction has too much unnecessary information and too much detailed. A lot of story information is available, must be shortened focusing on the brief information with previous studies, or lacks of the studies novelty of the present study after general information given.

Material and Methods must shortened selecting text or Table & Figures, not both. May be template questionnaire document could be given as appendix. Add the proper statistical test of the data regarding to groups, or factors what the authors tried to explain.

Results look like data report currently. Authors documented same information both in text and both Tables & Figures. It is recommended to keep Table and Figures, but not write or describe entire table & Figure in the text. This made text longer and hard-understanding especially with very long sentences. Microhabitats study is not clear-understood. 

Discussion and conclusion is too weak according to the results.

Overall, there is much info repeated along the manuscript few time in its different sections.

The sentences are really too long to follow, starting sentences, but it could be forgotten when we reach the end of the sentence. 

Regarding nice data of the present study, the MS could be revised to better present the study, so the MS needs a major revision. 

 Hard-understanding especially with very long sentences. 

The sentences are really too long to follow, starting sentences, but it could be forgotten when we reach the end of the sentence. 

Reviewer 3 Report

It is a study that evaluates the results of a survey.

I'm worried about the international attention of it.

In the material method section, information should be given about the content of the questionnaire.

Conclusion does not give clear information about the results

Reviewer 4 Report

A well-structured and well-written paper, with solid foundations and correctly described. To enrich the text, it is important to relate the size of the fish caught with the quantity taken from the rivers, it is important to inform the international reader of the consumption of each species. Aiming to clarify to the reader if the fish caught is consumed, where it demonstrates sustainability. The authors also put in the text hypotheses and possible scientific solutions based on the results.

Round 2

Reviewer 2 Report

The manuscript (MS) was better to understand the context.

The authors were indeed expected to apply statistical application as previously comments done by me. They could use percentage data for the statistical analyses. Without statistical results, however, the author presented highly claimed results in the MS.

There are some missing information between context of Material and Methods and results.

Line 174: please describe “upazila”, special to Bangladesh

Line 193, (See supplementary file)??, meaning that Questionnaire S1: Questionnaire for Fishers

Line 195-197, which statistics (mean and SD only?), no information results of statistical analyses in Results, or only graphs as the statistics?

No information how the authors categorize or classify vulnerability given in Table 2 which derived from a reference [13], which is not a result of the manuscript.

Data in Figure 5, could be statistically analyzed for a regional (spatial) difference.

Figure 6 did present the temporal distribution since data was derived from last 10-20 years. Add temporal comparison before and current data.  

In Table 4, how the authors estimate strength (etc. +, ++, +++) of negative impacts, method is missing in Mat and Met.

Line 520-522, “It is alarming to note that among the ten fish species that were frequently caught in the past 10-20 years, six of them are currently classified as threatened or near threatened in Bangladesh” How the authors conclude six species of them for the classification, please add category for this purpose into Mat and Met.

Overall, there are some points taken into attention to make clear information between results, Questionnaire questions and Mat and Med.

Reviewer 3 Report

manuscript can be accepted

Author Response

Thank you for accepting our revised manuscript.